# Mediterranean Diet, Vitamin D, and Hypercaloric, Hyperproteic Oral Supplements for Treating Sarcopenia in Patients with Heart Failure—A Randomized Clinical Trial

**DOI:** 10.3390/nu16010110

**Published:** 2023-12-28

**Authors:** Aura D. Herrera-Martínez, Concepción Muñoz Jiménez, José López Aguilera, Manuel Crespin Crespin, Gregorio Manzano García, María Ángeles Gálvez Moreno, Alfonso Calañas Continente, María José Molina Puerta

**Affiliations:** 1Maimonides Institute for Biomedical Research of Cordoba (IMIBIC), 14004 Córdoba, Spain; concepcion.munoz.sspa@juntadeandalucia.es (C.M.J.); jose.lopez.sspa@juntadeandalucia.es (J.L.A.); gregorio.manzano.sspa@juntadeandalucia.es (G.M.G.); mariaa.galvez.sspa@juntadeandalucia.es (M.Á.G.M.); contentine@gmail.com (A.C.C.); mariaj.molina.sspa@juntadeandalucia.es (M.J.M.P.); 2Endocrinology and Nutrition Service, Reina Sofia University Hospital, 14004 Córdoba, Spain; 3Cardiology Service, Reina Sofia University Hospital, 14004 Córdoba, Spain; manuel.crespin.sspa@juntadeandalucia.es

**Keywords:** oral supplements, heart failure, sarcopenia, LVEF, NT-proBNP

## Abstract

Background: Malnutrition and sarcopenia frequently affect patients with heart failure (HF), in which clinical outcomes and survival is decreased. Thus, appropriate nutritional screening and early nutrition support are highly recommended. Currently, nutritional support is not a standard of care in patients with HF, and the use of commercially available oral supplements (OSs) could provide an additional benefit to medical treatment in these patients. Aim: To compare the effect of the Mediterranean diet in combination with hypercaloric, hyperproteic OS in patients with HF. Patients and methods: An open label, controlled clinical study in which patients were randomly assigned to receive a Mediterranean diet (control group) vs. hypercaloric, hyperproteic OS (intervention group) for twenty-four weeks. Thirty-eight patients were included; epidemiological, clinical, anthropometric, ultrasound (muscle echography of the rectus femoris muscle of the quadriceps and abdominal adipose tissue), and biochemical evaluations were performed. All patients received additional supplementation with vitamin D. Results: Baseline malnutrition according to the GLIM criteria was observed in 30% of patients, while 65.8% presented with sarcopenia. Body cell mass, lean mass, and body mass increased in the intervention group (absolute increase of 0.5, *p* = 0.03, 1.2 kg, *p* = 0.03, and 0.1 kg, *p* = 0.03 respectively). In contrast, fat mass increased in the control group (4.5 kg, *p* = 0.05). According to the RF ultrasound, adipose tissue, muscle area, and circumference tended to decrease in the intervention group; it is probable that 24 weeks was too short a period of time for evaluating changes in muscle area or circumference, as previously observed in another group of patients. In contrast, functionality, determined by the up-and-go test, significantly improved in all patients (difference 12.6 s, *p* < 0.001), including the control (10 s improvement, *p* < 0.001) and the intervention group (improvement of 8.9 s, *p* < 0.001). Self-reported QoL significantly increased in all groups, from 68.7 ± 22.2 at baseline to 77.7 ± 18.7 (*p* = 0.01). When heart functionality was evaluated, LVEF increased in the whole cohort (38.7 ± 16.6 vs. 42.2 ± 8.9, *p* < 0.01); this increase was higher in the intervention group (34.2 ± 16.1 at baseline vs. 45.0% ± 17.0 after 24 weeks, *p* < 0.05). Serum values of NT-proBNP also significantly decreased in the whole cohort (*p* < 0.01), especially in the intervention group (*p* = 0.02). After adjusting by age and sex, nutritional support, baseline LVEF, NT-proBNP, and body composition parameters of functionality tests were not associated with mortality or new hospital admissions in this cohort. Conclusion: Nutritional support with hypercaloric, hyperproteic OS, Mediterranean diet, and vitamin D supplementation were associated with decreased NT-proBNP and improvements in LVEF, functionality, and quality of life in patients with HF, despite a significant decrease in hospital admissions.

## 1. Introduction

Nutritional support is essential in several human diseases [1]. Heart failure (HF) is a chronic disease with an increasing incidence (26 million people worldwide), especially in the elderly [2]. It represents the final stage of multiple cardiac diseases, significantly affects the quality of life (QoL) of patients, and is associated with elevated medical costs [3].

Sarcopenia is characterized by decreased muscle strength and fatigue due to reduced skeletal muscle mass, which is accompanied by muscle atrophy and decreased quality of muscle tissue. Specifically, muscle fibers are replaced by fibrotic tissue, which results in increased fragility and impaired muscle function [4]. Sarcopenia independently predicts worse clinical outcomes, increased morbidity and mortality in all patients [5,6], and, specifically, it has been directly associated with increased comorbidity in patients with HF and poorer clinical outcomes [7,8]. Furthermore, some authors consider that changes in muscle composition and synthesis are determining factors in the clinical evolution of HF, since muscle loss has been associated with poorer physical performance, higher oxygen consumption, increased hospitalization rates, decreased ventricular function, and lower survival [9]. In summary, sarcopenia significantly affects the prognosis of patients with HF [10].

Currently, there are no specific guidelines or recommendations concerning nutritional support in HF [11]. Most clinical societies support the idea that patients could follow a Mediterranean-style diet, probably due to the prevalence of anti-inflammatory nutrients, such as unsaturated fatty acids [12].

In the 1960s, the Seven Countries Study revealed that coronary heart disease and cancer rates were not elevated in a cohort of four Mediterranean areas: Crete and Corfu in Greece, Dalmatia in Croatia, and Montegiorgio in Italy. Specifically, at that moment, the Greek diet had the highest content of olive oil and was high in fruit, the Dalmatian diet was highest in fish, and the Italian diet was high in vegetables. These results represent the base of the Mediterranean diet, which recommends the consumption of plant food (fruits, vegetables, whole-grain cereals, nuts, and legumes); olive oil as the principal source of fat; moderate amounts of fish and poultry; and relatively low consumption of red meat [13]. According to the PREDIMED study, the Mediterranean diet provides protection against the development of cardiovascular diseases [14].

The European and American Societies of Enteral and Parenteral Nutrition state that oral nutritional supplements provide additional nutrients, including protein and energy, for people who are not meeting their nutrition needs through food alone, due to decreased appetite or food intake, increased nutrition needs, or poor absorption of nutrients due to illness. Thus, when oral feeding is insufficient, oral nutritional supplementation is indicated; specifically, hypercaloric (1.5–2 kcal/mL) and, if necessary, hyperproteic oral supplements (OS) are generally recommended, since they can improve inflammation status, QoL, and survival in HF patients [15].

Interestingly, a previous multicenter clinical trial has suggested that diet optimization, specific nutritional recommendations, and nutritional supplement prescriptions (when necessary) in patients with HF may have a prognostic benefit during and after hospital admission [16]. Despite these promising results, there are no clinical trials or studies that specifically evaluate the benefit of nutritional support in patients with HF [11].

Interestingly, it has been described that in patients with HF, left ventricular ejection fraction (LVEF) has a positive predictive value for adverse effects [7,8]. Additionally, recent studies have shown that myofibrillar atrophy and its functional alteration are specifically related to lower LVEF [9]. Thus, a strategy that allows stopping or improving muscle atrophy could improve LVEF, and, as a consequence, the outcomes of patients with HF. In this context, the aim of this study was to compare the clinical evolution of patients with HF (with moderate or reduced LVEF) when receiving a Mediterranean diet alone or in combination with a hypercaloric, hyperproteic OS for twenty-four weeks.

## 2. Materials and Methods

### 2.1. Patients

This study was approved by the Ethics Committee of the Reina Sofia University Hospital (Cordoba, Spain; reference number 5164 approved on 21 October 2021 and updated on 30 May 2023). It was conducted in accordance with the Declaration of Helsinki and according to national and international guidelines. This is a prospective open label study, wherein a written informed consent was signed by every individual before inclusion in the study. All patients received information before the inclusion and were included only if they agreed to participate. The inclusion criteria were patients of both sexes, age > 18 y-old < 85 y-old, LVEF < 50%, and a hospital admission due to HF in the previous 6 months. Exclusion criteria were end-stage kidney and/or liver disease. The sample size was calculated based on the usual number of admissions due to HF for 6 months in this hospital (specifically during the winter and spring seasons). Thirty-eight consecutive patients were included. A schematic overview of the study is depicted in Figure 1.

### 2.2. Study Design

This was a randomized, open, controlled, clinical trial. Patients were randomly assigned by the clinical investigator to receive either a Mediterranean diet alone or a Mediterranean diet plus two hypercaloric, hyperproteic OSs per day with a 1:1 allocation for twenty-four weeks. Specifically, the OS was composed with a mixture of slow-release carbohydrates and fiber; the specific composition of the OS is presented in Appendix A. OSs were kindly donated by Vegenat Healthcare^®^ (Badajoz, Spain). When included in the study, all patients received general education and advice about nutritional support, the Mediterranean diet, and physical activity; additionally, patients received oral supplementation with calcifediol (in different doses in order to reach levels of sufficiency, defined as a serum 25 OH vitamin D levels > 30 ng/dL). Fifty-six patients were assessed for eligibility; 14 did not met the inclusion criteria and 4 declined to participate; 38 patients were consecutively 1:1 randomized, i.e., 19 patients in each arm. Five patients died during the study period (four in the control arm and one in the intervention arm). The clinical trial design is depicted in Figure 2.

Nutritional evaluation was performed by the same endocrinologist in all cases. Physical examination included body composition analysis (bioelectrical bioimpedance, and abdominal, arm and calf circumferences), functional tests (up-and-go test and handgrip strength) and nutritional ultrasound of abdominal adipose tissue and rectus femoris (RF) muscle of the quadriceps. Specifically, vectorial bioelectrical bioimpedance (BIA) was performed using a NUTRILAB-Akern impedanciometer (Akern, Pis, Italy); for handgrip strength, a Jamar^®^ hydraulic dynamometer was used; and the nutritional echography was performed using a GE Logiq E9 ultrasound machine (General Electric, Madrid, Spain) and a linear 9L-D probe (Relmedic, Madrid, Spain). Additionally, a self-reported quality of life value using a 0–100 scale was collected.

### 2.3. Outcomes

The primary outcome was to evaluate the change in lean mass of patients after both nutritional interventions. Secondary outcomes included changes in other parameters of body composition (fat mass, water, bone, phase angle, body cell mass (BCME), extracellular mass (ECME)); anthropometric parameters (calf, arm, and abdominal circumference); LVEF assessed by transthoracic ultrasound, biochemical nutritional parameters (hemoglobin, lymphocytes, total cholesterol, total high-density lipoprotein (HDL) cholesterol, low-density lipoprotein (LDL) cholesterol, triglycerides, transferrin, ferritin, albumin, prealbumin *C*-reactive protein (C-RP), *N*-terminal pro-brain natriuretic peptide (NT-proBNP); functionality (stand-up test and handgrip strength); QoL; ultrasound-measured nutritional variables including: 1. Abdominal fat mass ultrasound (total abdominal adipose tissue, subcutaneous adipose tissue and pre-peritoneal fat) and 2. RF ultrasound (subcutaneous adipose tissue, RF-Y and RF-X axis, muscle area and circumference). This morphofunctional nutritional evaluation was performed as previously described [1,17,18]. The presence of malnutrition according to the GLIM criteria [19] and sarcopenia (defined as age- and gender-adjusted handgrip strength ≤ p25) was also determined.

### 2.4. Statistical Analysis

Between-group comparisons were analyzed using the Mann–Whitney U test (nonparametric data). Paired analysis was performed by Wilcoxon test (nonparametric data). The chi-squared test was used to compare categorical data. Statistical analyses were performed using SPSS statistical software version 20, and Graph Pad Prism version 6. Data are expressed as medians with interquartile ranges and percentages. Absolute differences in some parameters were calculated using mean values. For specific group analysis, the absolute number is also expressed in brackets. *P*-values < 0.05 were considered statistically significant.

## 3. Results

### 3.1. Baseline Characteristics of the Groups

Thirty-eight patients were included. Most of them were male (71.1%) with a median age of 67.5 y-old. Both groups were comparable in most clinical variables (Table 1). Specifically, patients in the control group tended to be older (72 (64.5–80)) than patients in the intervention group. Active smokers were more frequent in the intervention group (21.1%) than the control group (15.8%), while previous tobacco exposure was higher in the latter (42.1%) compared with the intervention group (5.3%, *p* < 0.01). No statistically significant differences were observed between groups regarding previous history of type 2 diabetes or ischemic cardiomyopathy, either when body weight, weight loss, or food intake were compared. Remarkably, patients in the intervention group tended to present with a higher rate of incomplete denture (78.9% vs. 47.4%, *p* = 0.05).

Significantly, there were no significant differences between baseline LVEF and NT-proBNP levels between both groups. Despite this, control patients tended to complain more frequently about abdominal pain (21.1% vs. 0%, *p* = 0.05) and dyspnea (94.5% vs. 63.2%, *p* = 0.09); they also tended to refer to a higher number of resting hours per day (9 vs. 7 h/day, *p* = 0.09) and their self-rated health score was significantly lower (60 vs. 76, *p* = 0.04). Detailed baseline clinical characteristics of both are depicted in Table 1.

Baseline malnutrition according to the GLIM criteria was observed only in 24% of patients, while 65.8% presented with sarcopenia. Specifically, most patients presented handgrip strength < p5 (28.9%), followed by p25 (23.1%), p50 (21.1%) and p10 (13.2%). Handgrip strength > p50 was observed only in 13.2% of the evaluated patients. There were no statistically significant differences in the distribution of sarcopenia and malnutrition in both groups, but sarcopenia tended to be higher in the control group (78.9% vs. 52.6%; *p* = 0.09).

Sarcopenia was associated with increased age and increased C-RP serum levels, but not with NT-proBNP levels or LVEF (Figure 3).

At baseline, NT-proBNP levels negatively correlated with some BIA parameters including body weight, BMI, BCME, ECME, and fat-, lean- and bone mass; anthropometric parameters including abdominal and arm circumference; both functional parameters, increased time in the up-and-go test and decreased handgrip strength, and muscle ultrasound parameters including RF area, circumference, X-axis; and adipose tissue. NT-proBNP also negatively correlated with hemoglobin, albumin, prealbumin, and triglycerides; in contrast, it positively correlated with C-RP serum levels. LVEF negatively correlated only with serum hemoglobin (Figure 4A).

Adherence to OS was assessed every 21 days. At the end of the study, 71.1% of patients took two supplements per day, 9.8% one per day, 1.9% one and a half per day, and 0.5% half an OS per day.

After twenty-four weeks of intervention, there were no significant differences between the prevalence of new hospital admissions, weight loss, or gastrointestinal symptoms; the number of resting hours still tended to be higher in the control group (median 8 vs. 4.5 h/day, *p* = 0.08) and the self-rated score was also significantly lower (median 75 vs. 85, *p* = 0.03). Detailed characteristics of both groups after the intervention are depicted in Table 2.

At the end of the study, NT-proBNP levels negatively correlated with lymphocytes, serum prealbumin, and triglycerides, while LVEF negatively correlated with RF X-axis in controls (Figure 4B). In the intervention group, serum NT-proBNP levels negatively correlated with BIA parameters including body weight, ECME, BCME, and lean and bone mass; anthropometric parameters including arm and calf circumference; functional parameters (up-and-go test and handgrip strength); and ultrasound variables (RF muscle area). LVEF correlated positively with C-RP and negatively with albumin levels. (Figure 4C).

### 3.2. Primary and Secondary Outcomes

Body weight significantly increased in the whole cohort at the end of the study (79 (69–85) vs. 81 (68–88) kg, *p* = 0.02). This change was clinically more significant (but not statistically significant) in the control group (increase of 7.2 kg; *p* = 0.18). In contrast, body weight tended to significantly increase in the intervention group (200 g, *p* = 0.06). As a consequence, BMI increased in the whole cohort (28.5 kg/m^2^ (25–30) at baseline vs. 28.5 (25.6–32.5) at the end of the study, *p* = 0.02) and was statistically significant in the intervention group (27.4 kg/m^2^ (23.5–28.9) at baseline vs. 27.4 kg/m^2^ (24–28.6) at the end of the study, *p* = 0.03). BCME tended to decrease in the control group (difference of 0.7 kg, *p* = 0.08) and significantly increased in the intervention group (increase of 0.5 kg, *p* = 0.03); in contrast, fat mass tended to increase in the whole cohort, especially in the control group (20.2 kg (14.6–25.8) at baseline vs. 25 kg (23.3–31.4) at the end of the study, *p* = 0.05). When lean mass was analyzed, it significantly increased in the intervention group (55.3 kg (50.1–61.1) at the end of the study vs. 52.5 kg (49.8–60.9) at baseline, *p* = 0.03). Additionally, bone mass also slightly increased in this group (difference: 0.1 kg, *p* = 0.03; Table 3).

No statistically significant differences were observed In abdominal, arm, or calf perimeters after 24 weeks of intervention (Table 3).

According to the RF ultrasound, adipose tissue, muscle area, and muscle circumference tended to decrease in the intervention group; specifically, adipose tissue decreased 0.1 cm (*p* = 0.08), muscle area 0.9 cm^2^ (*p* = 0.06), and muscle circumference 1 cm (*p* = 0.05). There were no differences in fat mass distribution according to the abdominal adipose tissue ultrasound (Table 3).

Regarding functionality tests, no significant change in handgrip strength was observed, but mobility significantly improved in all patients by 12.6 s according to the up-and-go test (*p* < 0.001) in both the control (10 s improvement, *p* < 0.001) and intervention groups (improvement of 8.9 s, *p* < 0.001; Table 3).

Regarding the biochemical data, hemoglobin increased by 0.4 mg/dL in all patients (*p* = 0.003), and especially in the intervention group (increase of 0.3 mg/dL *p* = 0.02); in contrast, ferritin significantly decreased in all patients (106 mg/dL (35–176) at baseline vs. 73 mg/dL (32–111) at the end of the study, *p* = 0.003), and especially in the intervention group (130 mg/dL (104–169) at baseline vs. 80 mg/dL (37–113) at the end of the study, *p* < 0.001). The visceral protein transferrin increased in the intervention group by 10 mg/dL at 24 weeks, (*p* = 0.04), while LDL cholesterol decreased 0.6 mg/dL in the same group (*p* = 0.04). A decrease in serum triglycerides was observed in all patients (136 mg/dL (101–174) vs. 112 mg/dL (91–147), *p* < 0.001), and especially in the control group (144 mg/dL (92–209) vs. 117 mg/dL (102–152), *p* = 0.01). Additionally, C-RP significantly decreased in all patients (8.5 mg/L ± 15.0 vs. 2.8 mg/L ± 4.8, *p* = 0.02), and especially in the intervention group (decreased by 5 mg/L, *p* < 0.01). Due to treatment with calcifediol, 25(OH)D levels tended to increase in all cases (absolute increase of 4.5 ng/dL, *p* = 0.08), and especially in patients that received OS (7.3 ng/dL, *p* = 0.08; Table 4).

Self-reported QoL significantly increased in all patients, from 72.5 (50–85) at baseline to 80 (80–90) (*p* = 0.01). In the control group, this change was not significantly different (65 (40–78) at baseline vs. 75 (67–80) after 24 weeks, *p* = 0.10); in contrast, in the intervention group, QoL tended to increased (80 (70–90) vs. 85 (75–95); *p* = 0.07).

When heart functionality was evaluated, LVEF increased in the whole cohort (38.7% ± 16.6 vs. 42.2% ± 8.9, *p* < 0.01; Figure 3); this increase was higher in the intervention group (34.2% ± 16.1 at baseline vs. 45.0% ± 17.0 after 24 weeks, *p* < 0.05) than in the control group (43.2% ±16.3 vs. 48.2% ± 8.9). Serum values of NT-proBNP also significantly decreased in the whole cohort (1855 pg/mL (393–4364) at baseline and 741 pg/mL (393–1992) after 24 weeks, *p* < 0.01). In particular, the intervention group (1952 pg/mL (1179–3307 at baseline) decreased by 1303 pg/mL (741–2111) after 24 weeks (*p* = 0.02). Although NT-proBNP also decreased in the control group, this change was not statistically significant (Table 4, Figure 5).

Age- and sex-adjusted analysis revealed that nutritional support, baseline LVEF, NT-proBNP, body composition parameters, and functionality tests were not associated with mortality or new hospital admissions in this cohort (some representative parameters are depicted in Table 5).

## 4. Discussion

The European Society of Cardiology defines HF as “a clinical syndrome characterized by typical symptoms (e.g., breathlessness, ankle swelling and fatigue) that may be accompanied by signs (e.g., elevated jugular venous pressure, pulmonary crackles and peripheral edema) caused by a structural and/or functional cardiac abnormality, resulting in a reduced cardiac output and/or elevated intra-cardiac pressures at rest or during stress” [20]. The prevalence of HF is 2% of the population in developed countries, and HF is associated with increased morbidity, institutionalization, and mortality [20]. HF especially affects elderly people, and its increase is probably related to the availability of therapeutic strategies that have improved life expectancy in individuals with HF [10].

These patients frequently present malnutrition, within a range of 10–50%, depending on the stage of the disease and according to the parameters used to define malnutrition. In patients with advanced HF, it can progress to cardiac cachexia. Both malnutrition and cachexia have been associated with worse prognosis, period of hospital stay, readmission rate, in-hospital mortality, and overall mortality [21,22]. In these cases, nutritional support is highly recommended. In a Spanish study that included 304 patients treated in a HF unit, mortality in malnourished patients (classified using the Mini Nutritional Assessment (MNA) screening test) was higher (68.9%) compared to those who were at risk of malnutrition (33.3%) or to non-malnourished patients (15.2%; *p* < 0.001); additionally, malnutrition was an independent predictor of mortality [23]. In another Spanish study performed in 150 ambulatory patients, the presence of malnutrition according to the MNA score was an independent predictor of all-cause mortality, cardiovascular mortality, and HF admissions, even if patients did not present malnutrition according to GLIM criteria [13]. Consistent with this, we observed a baseline prevalence of malnutrition of 24% using the GLIM criteria. In contrast, 65.8% presented sarcopenia, suggesting that GLIM criteria allow the diagnosis of advanced malnutrition in this population and other tools should be used to evaluate this population.

Additionally, some patients do not lose weight, despite presenting with overweight or obesity accompanied with sarcopenia. Thus, muscle mass is decreased in quality, quantity, and functionality independently of body weight [10]. In our cohort, 73.7% of the evaluated patients presented with obesity or overweight; thus, a significant proportion suffered sarcopenic obesity. Despite their body weight, these patients also require nutritional support [11]. Based on this, current nutritional evaluation should include parameters that appropriately evaluate muscle mass and functionality [17,24]. For these reasons, this study included a comprehensive nutritional evaluation.

A prevalence of sarcopenia of 34% has been previously reported in HF, and is higher in males (37% vs. 33% in females) [25]. We observed a higher rate of sarcopenia in this cohort. These differences might be related to the fact that this study only included patients of intermediate or decreased LVEF. In other series, sarcopenia was associated with older age, lower BMI, and higher NT-proBNP levels [25]. In contrast, in our study, sarcopenia was associated with older age, but not with serum NT-proBNP or LVEF; differences might be related to the method used to define sarcopenia and the sample size. Additionally, some studies have shown that myofibrillar atrophy and its functional alteration are related to a lower LVEF. Thus, a direct relation between sarcopenia and heart functionality has been suggested [26]. Regarding this point, we did not observe specific relations between LVEF and sarcopenia.

According to the Nutritional Intervention Program in Malnourished Patients Admitted for Heart Failure (PICNIC study), individualized nutritional intervention during and after HF admission may have a prognostic benefit. This multicenter clinical trial included 120 malnourished hospitalized patients with acute HF. Patients were randomized to conventional treatment or nutritional intervention (diet optimization, specific recommendations, and nutritional supplement prescriptions if necessary). At the end of the study, all-cause mortality or readmission for worsening of HF was significantly lower in the intervention group (27.1%) compared with the control group (60.7%) [16]. We did not observe differences in mortality of admissions, but, in contrast to the PICNIC study, in our cohort all patients received nutritional support with the Mediterranean diet with or without the OS. Despite the remarkable results of the PICNIC study, currently there are no standardized recommendations for nutritional support in patients with HF.

In this context, most experts and cardiology societies recommend a Mediterranean-style diet in these patients, probably due to its anti-inflammatory properties [12]. Additionally, according to the PREDIMED study, the Mediterranean diet provides protection against the development of cardiovascular diseases [14]. In our study, we observed that serum levels of NT-proBNP decreased, while LVEF increased in both groups, but more significant changes occurred in the intervention group. Although we can not attribute this clinical improvement only to the nutritional support, the effect of a general nutritional intervention in these patients combined with a rehabilitation program is remarkable.

Generally, it is recommended that hyperproteic OSs could be administered in patients with HF and insufficient oral intake [15]. In our study, OSs were administered to patients with and without appetite loss, and interestingly, a clinical benefit was observed in all patients, reflecting the importance of an appropriate nutritional support in HF independently of body weight or appetite. Importantly, several nutritional formulas of OSs are commercially available. Currently, there is no consensus about the type of supplement that should be administered; moreover, there is no evidence about the use of specific supplementation. Regarding this, β-hydroxy-β-methylbutyrate (HMB) is associated with improvement in nutritional status and a significant reduction in the risk of mortality compared with placebo in hospitalized patients for cardiovascular and pulmonary events such as congestive HF [27]; however, a specific study in HF patients has not been performed. In this context, we decided to use a hypercaloric, hyperproteic OS enriched with omega-3 and omega-6. Regarding this point, the European Society of Cardiology, the American College of Cardiology, and the American Heart Association state that supplementation with omega-3 fatty acids could be considered in these patients, since it has been suggested that their consumption reduces the risk of cardiovascular hospitalization and death in patients with HF (level of evidence B recommendation) [20,28]. This recommendation is based on the results of the GISSI-HF trial, which randomized 955 patients to receive a daily dose of 1 g of omega-3 polyunsaturated fatty acid (PUFA) or placebo, and showed that all-cause mortality and the combined end point of all-cause mortality or hospital admission for cardiovascular reasons were lower in the group receiving PUFA [29].

It is known that patients with HF have elevated circulating levels of pro-inflammatory cytokines; specifically, the innate and adaptive immune systems are activated in HF. Thus, treatment strategies targeting inflammation are a matter of interest in these patients [30]. As expected in our cohort, we observed elevated C-RP serum levels at baseline, reflecting this inflammatory status. These levels decreased in both groups after the intervention. This effect could have been related to the anti-inflammatory properties of the Mediterranean diet in combination with treatment adjustment and rehabilitation. Remarkably, this decrease was more evident in the intervention group, which could have been related in part to the presence of omega-3 in the OS composition, but also to the presence of slow-release carbohydrates in its formula. Specifically, slow-release carbohydrates are associated with lower postprandial glycaemia due to slower glucose release, lower postprandial insulin release, and stimulation of gut hormones [31]. In HF, a significant glucose peak after the meal intake should be avoided, since hyperglycemia is associated with disordered endothelial function, increased low-grade inflammation, increased blood coagulation, reduced fibrinolysis, decreased plaque stability, reduced triglyceride-rich lipoprotein and LDL removal, increased HDL cholesterol catabolism, reductions in free fatty acids, reduced early-phase insulin secretion, and increased insulin resistance, which are related to worse clinical outcomes [32,33].

In clinical nutrition, some nutritional formulas are recommended in specific situations; for example, immunonutrition is currently a standard of care in fast-track abdominal surgery programs, or after surgery in patients with head and neck cancer [18,34,35]. Based on the results of this study (significant decrease in ferritin and C-RP in the intervention group), it would be interesting to evaluate the effect of specific immunomodulatory OSs in patients with HF.

This study has some limitations: first the number of participants in each group, which can limit the findings of this study; furthermore, some baseline variables tended to be different in both groups despite there being no statistically significant differences. Ideally, a study with matched controls should be performed, considering that it would not reflect the regular clinical practice. Additionally, we did not supervise the diet daily intake of the patients. Finally, the nutritional intervention was not evaluated alone, which is a frequent limitation in studies regarding nutrition; it was not possible to evaluate the nutritional intervention in an isolated manner, since rehabilitation is currently considered a standard of care in patients with HF [36]. Thus, it would not be appropriate to exclude this intervention in the clinical management of patients with HF. It would be interesting to also determine long-term clinical response (for example, after 12 or 24 months). In contrast, this study has several strengths: first of all, the duration of the study; patients were evaluated after 24 weeks, while most studies of nutritional interventions evaluate patients for a period of between 6 and 12 weeks [1,37]. Additionally, the adherence to the OS was assessed and elevated in the intervention group and vitamin D supplementation was administered in both groups. This vitamin is frequently added (in different proportions) to OSs; thus, its use to achieve sufficiency levels in both groups was used to avoid confusing results. Regarding this point, although decreased vitamin D levels have been associated with worse clinical outcomes in several kinds of patients, including those with HF, its supplementation has not been associated with improved clinical evolution in patients with HF [38] or other endocrine-related diseases [39,40]. Furthermore, a comprehensive nutritional evaluation was performed including anthropometric, echography, functional, and biochemical parameters. Finally, to the best of our knowledge, this is the first clinical trial that specifically compared the clinical evolution in ambulatory patients with HF after receiving two different strategies of nutritional support.

## 5. Conclusions

Taken together, our results reveal that a nutritional intervention with the Mediterranean diet in patients with HF results in improvement of functionality, QoL, and cardiac function. Furthermore, the use of a hypercaloric, hyperproteic OS (with slow-release carbohydrates, omega-3, and omega-6) is associated with lean mass gain, cell mass gain, improvement in biochemical nutritional parameters, and a more significant improvement in functionality, QoL, and LVEF, and a decrease in NT-proBNP serum levels, after 24 weeks of intervention. Regular recommendations of the Mediterranean diet should be provided to all patients after HF admission. Furthermore, malnutrition and sarcopenia screening should be regularly implemented before discharge, in order to recommend the use of OSs in patients with (or at risk of) malnutrition or sarcopenia. Additional head-to-head studies, with a larger sample size and comparing different OS formulas, should be performed.

## Figures and Tables

**Figure 1 nutrients-16-00110-f001:**
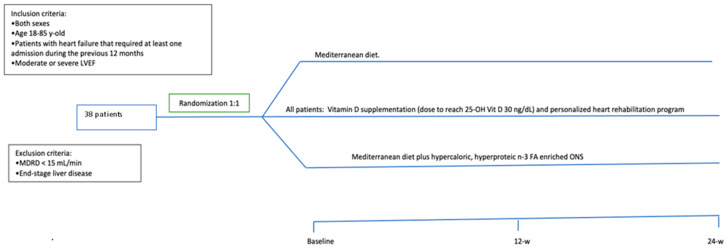
Schematic overview of the study and treatment arms.

**Figure 2 nutrients-16-00110-f002:**
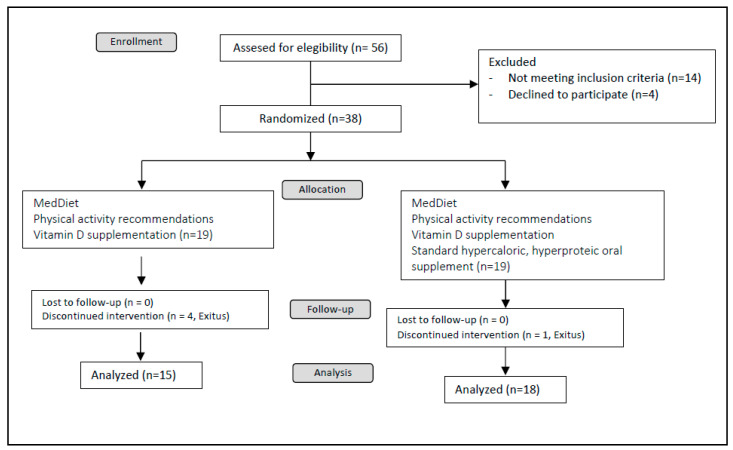
Study design.

**Figure 3 nutrients-16-00110-f003:**
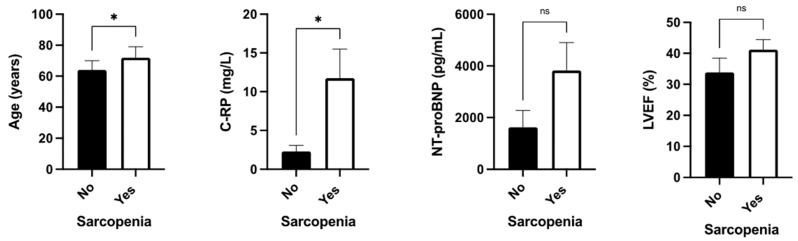
Clinical associations between sarcopenia with age, serum C-RP levels, serum NT-proBNP, and LVEF determined by transthoracic ultrasound. Legend: C-RP: c-reactive protein; ns: non-significative; *: *p* < 0.05.

**Figure 4 nutrients-16-00110-f004:**
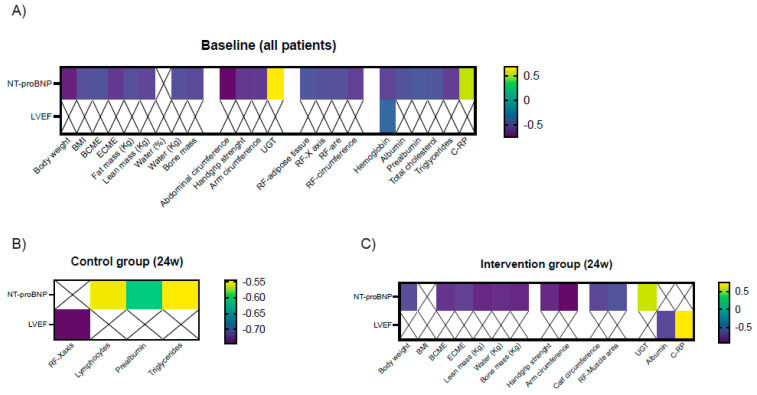
Significant clinical correlations between serum NT-proBNP levels and LVEF and nutritional parameters at baseline (**A**) and after 24 weeks in the control group (**B**) and in the intervention group (**C**). Legend: BMI: body mass index; BCME: body cell mass; ECME: extracellular mass; UGT: up-and-go test; RF: rectus femoris; C-RP: C-reactive protein.

**Figure 5 nutrients-16-00110-f005:**
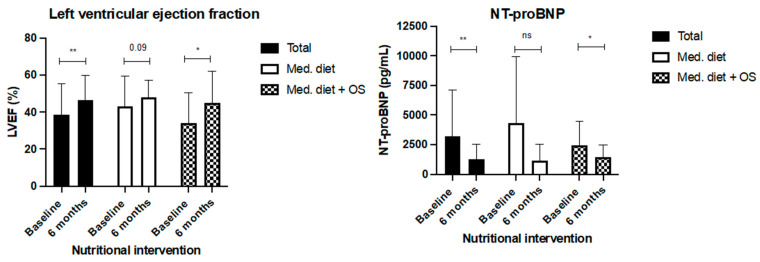
Change in LVEF and serum NT-pro BNP serum levels after 24 weeks of nutritional intervention with Mediterranean diet (control group) and Mediterranean diet plus two hypercaloric, hyperproteic oral supplements (Oss). Legend: *: *p* < 0.05; **: *p* < 0.01; ns: non-significant.

**Table 1 nutrients-16-00110-t001:** Baseline clinical characteristics of the patients. Comparison between groups based on the nutritional intervention.

Characteristics	Total (*n* = 38)	Mediterranean Diet (*n* = 19)	Mediterranean Diet and OS (*n* = 19)	*p*
Sex (♂/♀)	71.1%/28.9% (11/27)	31.6/68.4 (6/13)	73.7/26.3 (14/5)	0.50
Age (years)	67.5 (61–78)	72 (64.5–80)	65 (56–72)	0.06
Tobacco exposure (%)				0.01
No	57.9 (22/38)	42.1 (8/19)	73.7 (14/19)	
Active	18.4 (7/38)	15.8 (3/19)	21.1 (4/19)	
Previous exposure	23.7 (9/38)	42.1 (8/19)	5.3 (1/19)	
Type 2 Diabetes	42.1 (16/38)	36.8 (7/19)	47.4 (9/19)	0.38
Previous ischaemic cardiomyopathy	34.2 (13/38)	36.8 (7/19)	31.6 (6/19)	0.50
Ejection fraction (%)	33 (25–49.5)	40 (32.5–54)	38 (23–35)	0.46
NT-proBNP (pg/mL)	4225 (2001–7289)	3678 (1966–7203)	4412 (2177–7255)	0.59
Current weight (kg)	78 ± (70.3–89.5)	81 (75–90)	76 (70–85)	0.17
Symptoms (%)				
Weight loss (3 months)	55.3 (21/38)	47.4 (9/19)	63.2 (12/19)	0.26
Weight loss kg (3 months)	2 (0–4.75)	0 (0–3.5)	2 (0–5)	0.56
Weight loss (6 months)	28.9	31.6 (6/19)	26.3 (5/19)	0.50
Weight loss kg (6 months)	0.5 (0–3)	0.1 (0–3)	0 (0–1.5)	0.80
Uncomplete denture	63.2 (24/38)	47.4 (9/19)	78.9 (15/19)	0.05
Food intake (%)				
Soft	7.9 (3/38)	5.3 (1/19)	10.5 /2/19)	0.50
Normal	92.1 (35/38)	94.7 (18/19)	89.5 (17/19)	0.50
Gastrointestinal symptoms (%)	15.8 (6/38)	15.8 (3/19)	15.8 (3/19)	0.70
Abdominal pain	10.5 (4/38)	21.1 (4/19)	0	0.05
Nauseas/vomits	5.3 (2/38)	0	10.5 (2/19)	0.24
Diarrhea	5.3 (2/38)	0	10.5 (2/19)	0.24
Body lesions	0	0	0	-
Dyspnea	78.9 (30/38)	94.7 (18/19)	63.2 (12/19)	0.09
Malnutrition (%)	23.7 (9/38)	26.3 (5/19)	21.1 (4/19)	0.50
Sarcopenia (%)	65.8 (25/38)	78.9 (15/19)	52.6 (10/19)	0.09
Physical activity (%)				
Intense	0	0	0	-
Moderate	18.4 (7/38)	10.5 (2/19)	26.3 (5/19)	0.20
Resting time (hours/day)	7 (5–10)	10 (6–11)	6 (4–8.5)	0.09
Quality of life				
Self-rated health score	72.5 (50–85)	65 (40–78)	80 (70–90)	0.04
Overweight/obesity (%)	73.7 (28/38)	57.1 (16/19)	42.9 (12/19)	0.14

Legend: Categorical data are presented as percentages and the absolute number in brackets.

**Table 2 nutrients-16-00110-t002:** Clinical characteristics of the patients twenty-four weeks after nutritional support.

Characteristics	Total (*n* = 32)	Mediterranean Diet (*n* = 15)	Mediterranean Diet and OS (*n* = 18)	*p*
Heart failure hospitalizations (%)	29 (9/31)	33.3 (5/15)	22.2 (4/18)	0.37
Number of hospitalizations	0.5	0.5	0.5	0.71
Weight loss (%)	21.2 (7/32)	20 (3/15)	22.2 (4/18)	0.58
Gastrointestinal symptoms (%)	15.2 (5/32)	20 (3/15)	11.1 (2/18)	0.44
Abdominal pain	6.3 (2/32)	13.3 (2/15)	0	0.21
Nauseas/vomits	3.1 (1/32)	0	5.5 (1/18)	0.53
Diarrhea	7.9 (3/32)	6.7 (1/15)	11.8 (2/18)	0.55
Body lesions	0	6.7 (1/15)	0	-
Dyspnea	75 (26/32)	93.7 (14/15)	58.8 (10/18)	0.09
Physical activity (%)				
Intense	0	0	0	-
Moderate	34.4 (11/32)	26.7 (4/15)	41.2 (7/17)	0.31
Resting time (hours)	6 (4–10)	8 (6–10)	4.5 (4–7.3 )	0.08
Quality of life	80 (80–90)	75 (67–80)	85 (75–95)	0.03
Mortality (%)	13.2 (5/38)	21.1 (4/19)	5.3 (1/19)	0.17

Legend: Categorical data are presented in percentages and the absolute number in brackets.

**Table 3 nutrients-16-00110-t003:** Morphofunctional assessment of the nutritional status at baseline and six months after nutritional treatment.

	Total			Mediterranean Diet			Mediterranean Diet and OS		
Characteristics	Baseline (*n* = 38)	Six Months (*n* = 33)	p1	Baseline (*n* = 19)	Six Months (*n* = 15)	p2	Baseline (*n* = 19)	Six Months (*n* = 18)	p3
Body weight	79 (69–85)	81 (68–88)	0.02	82.5 (73.9–90.1)	88.1 (81.3–92.7)	0.18	78.6 (65.1–81.9)	69.8 (65–84)	0.06
Bioimpedance analysis									
BMI (kg/m^2^)	28.5 (25–30)	28.5 (25.6–32.5)	0.02	29.6 (27–34.3)	30.6 (28.2–36.1)	0.21	27.4 (23.5–28.9)	27.4 (24– 28.6)	0.05
BCME (kg)	37.2 (33.3–43)	37 (33–42.1)	0.67	38.1 (33.8–43.8)	36.9 (33.3–42)	0.08	35.5 (33.4–41.2)	38.2 (33.3–42.2)	0.03
ECMe (kg)	20.7 (19.8–22.8)	21.7 (19.3–22.9)	0.24	21.5 (20–22.9)	22.5 (20.1–23.9)	0.42	20.4 (18.9–22.3)	21.2 (18.1–22.7)	0.36
Fat mass (%)	25.1 (21.7–31.5)	25 (23.3–31.4)	0.19	27.8 (23.2–33.3)	30.9 (27.5–30.1)	0.16	23.4 (20.6–23.3)	23.5 (28.8–24.9)	0.82
Fat mass (kg)	20.2 (14.6–25.8)	25 (23.3–31.4)	0.07	23.2 (17.4–27.6)	26.7 (22.1–30.1)	0.05	17.5 (13–22.4)	15.4 (12.5–19.7)	0.48
Lean mass (%)	71.2 (65.1–74.2)	71.3 (65.2–72.9)	0.35	68.6 (63.5–73)	65.7 (61.7–68.9)	0.20	72.8 (70.1–75.4)	72.7 (70.1–75.4)	0.82
Lean mass (kg)	52.6 (50.2–61.3)	55.6 (50.1–62.3)	0.30	56.6 (50.5–62)	55.7 (50.1–62.5)	0.30	52.5 (49.8–60.9)	55.3 (50.1–61.1)	0.03
Water (%)	52.6 (48.2–55.5)	52.8 (48.5–54.4)	0.43	51 (47.3–53.7)	48.6 (44.9–51.6)	0.22	54.7 (51.6–57)	54.1 (52.9–56.9)	0.83
Water (kg)	40.1 (36.8–45.5)	42.4 (35.9–45.8)	0.40	40.7 (37.3–46.1)	42.5 (35.9–46.1)	0.38	39.2 (36.5–44.9)	42.3 (37–45.5)	0.12
Bone Mass (kg)	2.8 (2.7–3.2)	2.9 (2.7–3.2)	0.52	3 (2.7–3.3)	2.9 (2.7–3.3)	0.14	2.6 (2.6–3.2)	2.9 (2.7–3.2)	0.03
Phase angle	4.7 (3.7–6)	4.5 (3.6–5.9)	0.80	3.7 (3.1–5.5)	4.5 (3.3–4.9)	0.53	4.9 (4.5–6.2)	4.8 (4.3–6.5)	0.46
Anthropometric evaluation									
Abdominal circumference	106 (98–110)	106 (93–113)	0.90	110 (102.5–114)	112 (107–121)	0.59	101 (93–108)	104 (92–106)	0.62
Arm circumference	32 (28–33)	31 (28–33)	0.49	32 (28–33)	32 (30–35)	0.14	31 (28–33)	30 (28–32)	0.62
Calf circumference	37 (34–39)	37 (35–39)	0.53	37 (35–39)	38 (36–41)	0.58	37 (34–40)	37 (35–38)	0.72
RF Muscle Ultrasound									
Adipose tissue (cm)	0.67 (0.5–0.8)	0.61 (0.5–1)	0.53	0.69 (0.5–0.8)	0.8 (0.6–1.2)	0.53	0.69 (0.5–0.8)	0.5 (0.5–0.7)	0.08
Area (cm^2^)	3.75 (2.2–4.6)	3.1 (2.6–3.7)	0.19	3.1 (2.1–4.6)	3.3 (2.7–3.7)	0.97	3.5 (2.7–4.4)	3.0 (2.6–3.5)	0.06
Circumference (cm)	8.7 (7.5–10.1)	8.2 (7.8–9.2)	0.42	8.3 (6.8–9.9)	9.0 (7.8–9.5)	0.46	9.5 (7.9–10.1)	8.1 (7.8–9.2)	0.05
AP axis (cm)	1.2 (1–1.4)	1.1 (0.9–1.2)	0.20	1.3 (0.9–4.6)	1.2 (1.0–1.3)	0.90	1.2 (1.0–1.5)	1.0 (0.9–1.2)	0.09
Transversal axis (cm)	3.8 (3.3–4.4)	3.3 (3.6–4)	0.42	3.8 (3.1–4.5)	3.6 (3.3–4.2)	0.78	3.7 (3.4–4.2)	3.7 (3.4–4.2)	0.30
Abdominal Ultrasound									
Total adipose tissue (cm)	2.2 (1.5–3)	2.3 (1.6–3.1)	0.87	2.4 (3.4–4.2)	2.3 (1.6–3.8)	0.78	1.7 (1.5–3.0)	2.2 (1.6–3.0)	0.87
Subcutaneous adipose tissue (cm)	1.6 (1–1.3)	1.6 (1–2)	0.53	1.6 (0.9–2.1)	1.6 (1.0–2.1)	0.20	1.6 (1.1–2.5)	1.6 (1.0–1.9)	0.92
Preperitoneal fat (cm)	0.5 (0.4–0.7)	0.6 (0.4–0.7)	0.37	0.5 (0.4–0.7)	0.5 (0.4–0.7)	0.71	0.5 (0.4–0.8)	0.6 (0.5–0.7)	0.39
Functional evaluation									
Handgrip strenght (dominant arm, kg)	30 (20.3–39.8)	31.7 (18–40.8)	0.71	23 (19–24)	24 (13–36)	0.36	34 (29–40)	38 (25–44)	0.44
Up-and-go test (seconds)	20 (16–25)	11 (9.5–13.7)	<0.001	20 (18–25)	11 (10–16)	<0.001	18 (14–21)	10 (9–12)	<0.001

Legend: RF: rectus femoris. p1 refers to the comparison between all patients at baseline and after twenty-four weeks; p2 refers to the comparison between patients of the control group (Mediterranean diet) at baseline and after twenty-four weeks; p3 refers to the comparison between patients of the intervention group (Mediterranean diet plus oral nutritional supplementation) at baseline and after twenty-four weeks.

**Table 4 nutrients-16-00110-t004:** Biochemical analysis at baseline and six months after nutritional support.

	Total			Mediterranean Diet			Mediterranean Diet and OS		
Characteristics	Baseline (*n* = 38)	Six Months (*n* = 33)	p1	Baseline (*n* = 19)	Six Months (*n* = 15)	p2	Baseline (*n* = 19)	Six Months (*n* = 18)	p3
Biochemical parameters									
Hemoglobin (g/dL)	13.8 (12.9–15.3)	14.2 (13–15.6)	0.03	13.5 (12.7–15)	14 (13–14.8)	0.22	13.9 (13.1–15.5)	14.9 (13.2–15.7)	0.03
Lymphocytes (x mm^3^)	1730 (1320–2360)	1655 (1324–1912)	0.13	1730 (1470–2150)	1860 (1390–2060)	0.41	1665 (1267–2420)	1550 (1070–1810)	0.18
Albumin (g/dL)	4.5 (4.3–4.9)	4.5 (4.2–4.7)	0.30	4.5 (4.4–4.9)	4.6 (4.3–4.8)	0.81	4.5 (4.3–4.9)	4.4 (4.2–4.6)	0.28
Prealbumin (mg/dL)	26 (21–28)	25 (22–28)	0.55	25 (20–28)	24 (21–28)	0.76	27 (23–29)	25 (22–28)	0.23
Ferritin (mg/dL)	106 (35–176)	73 (32–111)	0.003	74 (32–171)	80 (37–113)	0.46	130 (104–169)	80 (37–113)	<0.01
Transferrin (mg/dL)	245 (218–262)	239 (227–286)	0.12	252 (233–308)	240 (232–299)	0.71	240 (214–253)	234 (223–285)	0.04
Total cholesterol (mg/dL)	158 (126–196)	161 (133–192)	0.72	159 (126–187)	166 (131–195)	0.44	157 (126–195)	157 (141–190)	0.11
HDL cholesterol (mg/dL)	46 (41–53)	45 (38–51)	0.17	44 (39–50)	44 (32–50)	0.11	48 (45–54)	47 (38–56)	0.85
LDL cholesterol (mg/dL)	86 (65–115)	93 (61–123)	0.12	80 (55–103)	99 (58–120)	0.78	86 (70–122)	90 (61–123)	0.04
Triglycerides (mg/dL)	136 (101–174)	112 (91–147)	<0.001	144 (92–209)	117 (102–152)	0.01	132 (105–164)	90 (61–123)	0.17
C-RP (mg/L)	2.1 (0.5–6.9)	1.0 (0.5–2.6)	0.02	2.2 (0.5–15)	2.1 (0.5–5.6)	0.79	1.4 (0.7–5.8)	0.7 (0.5–1.5)	<0.01
NT-proBNP (pg/mL)	1855 (1080–4364)	741 (393–1992)	<0.01	1757 (557–6027)	489 (178–1676)	0.17	1952 (1179–3307)	1303 (741–2111)	0.02
Vitamin D (ng/dL)	18 (11–26)	22 (16–31)	0.08	15 (11–21)	17 (9–29)	0.51	17 (9–29)	25 (19–38)	0.08

p1 refers to the comparison between all patients at baseline and after twenty-four weeks; p2 refers to the comparison between patients of the control group (Mediterranean diet) at baseline and after twenty-four weeks; p3 refers to the comparison between patients of the intervention group (Mediterranean diet plus oral nutritional supplementation) at baseline and after twenty-four weeks.

**Table 5 nutrients-16-00110-t005:** Multivariate logistic regression for mortality and new hospital admissions in patients with HF that received nutritional support after adjusting by age and sex.

Variable		OR	CI	*p*
Mortality	Nutritional support	0.24	0.02–2.62	0.24
	Baseline LVEF	0.98	0.92–1.05	0.64
	Baseline NT-proBNP	1.00	1.00–1.00	0.67
	Baseline BMI	0.94	0.75–1.18	0.61
	Baseline BCME	0.99	0.90–1.10	0.88
	Baseline Phase angle	0.51	0.19–1.37	0.18
	Baseline Handgrip strength	0.92	0.81–1.05	0.22
	Baseline up-and-go test	1.09	0.98–1.23	0.11
	Baseline Preperitoneal fat	0.36	0.00–15.48	0.60
	Baseline C-RP	0.99	0.94–1.06	0.98
New hospital admissions	Nutritional support	0.65	0.12–3.57	0.62
	Baseline LVEF	1.00	0.95–1.05	0.94
	Baseline NT-proBNP	1.00	1.00–1.00	0.09
	Baseline BMI	1.06	0.89–1.26	0.48
	Baseline BCME	1.00	0.97–1.03	0.98
	Baseline Phase angle	0.92	0.63–1.32	0.63
	Baseline Handgrip strength	1.02	0.93–1.13	0.62
	Baseline up-and-go test	1.05	0.94–1.17	0.38
	Baseline Preperitoneal fat	0.25	0.01–4.38	0.34
	Baseline C-RP	1.01	0.97–1.06	0.48

## Data Availability

Data are contained within the article.

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
