# Peer review of "Mediterranean Diet, Vitamin D, and Hypercaloric, Hyperproteic Oral Supplements for Treating Sarcopenia in Patients with Heart Failure—A Randomized Clinical Trial"

_nutrients, 2023, doi:10.3390/nu16010110_

Round 1
Reviewer 1 Report
Comments and Suggestions for Authors
The article refers to two different dietetic interventions in regards of heart failure and sarcopenia.
The authors may consider following comments:
- The title does not reflect the importance of the study that was related to sarcopenia. It should be rewritten in such a way highlighting it.
- Study design may be somehow erroneous. It would be difficult to assess on how strong the effect of dietetics intervention, if it is concomitantly rehabilitation program and oral supplementation of D3 was started.
- Critical point may be the number of subjects. Drawing any conclusions from such a small study may be difficult, if not possible.
- Table 1. The qualitative data should be presented as a raw number and percent in the brackets the sign of “%” appears only in sex columns, the rest of data is not clear on what is representing.
- Last row of the table 1- there is an overlapping of text.
- Line 205- please explain what “self-reported core” means? I assume it refers to quality of life, but it is not clear.
- Title of table 2- “ weeks0”
- Last raw of table 2- there is an overlapping of text.
- Table 1 ,2, 3,4. Data is such small group barely can met criteria of normal distribution and homogeneity of variance. Hence it should be presented as median and lower upper quartiles. i.e. if we consider CRP in Total baseline group from table 4 ->8.5±15.0 it means that some study subjects presented negative level of CRP. Similarly NT pro-BNP.
- Table 4- Please compare data from Total baseline and Six-months. There is no correspondence, the numbers do not represent values of laboratory tests.
- Figure 5. What the stars refer to?
- Discussion part Only lines 328-330 mention the study results. The discussion part should be a trial of assessing obtained results through the prism of the cited papers. Here, the authors do not mention their results citing only the related papers.
Summary: As promising as this article initially seemed, it should be rejected in its current form. The problem is not only the extremely low number of tested groups, but also wrong presentation of statistical data, lack of description of the statistics used in the methodology, errors in data presentation (Table 4) and also the lack of discussion regarding the results obtained.
I suggest rethinking the entire study, correcting the above-mentioned errors and resubmission. However, the topic is promising.
Comments on the Quality of English Language-
Author Response
We sincerely thank the Reviewer for the constructive comments, which we found very helpful towards improving the quality of our study. Accordingly, specific changes have been made in the manuscript, based on these comments, as it is described in detail below in a point-by-point description of the changes introduced, and on how Reviewer’s concerns were addressed. Changes in the manuscript are indicated in red
REVIEWER 1
The article refers to two different dietetic interventions in regards of heart failure and sarcopenia.
The authors may consider following comments:
Reviewer: The title does not reflect the importance of the study that was related to sarcopenia. It should be rewritten in such a way highlighting it.
Authors: The article title has been adjusted following the reviewer´s suggestion
Reviewer: Study design may be somehow erroneous. It would be difficult to assess on how strong the effect of dietetics intervention, if it is concomitantly rehabilitation program and oral supplementation of D3 was started.
Authors: This is a frequent limitation in studies regarding nutrition, it is not possible to evaluate the nutritional intervention in an isolated manner, currently rehabilitation is considered a standard of care in patients with HF, thus, it would not be ethic to exclude this intervention in the clinical management of these patients. Additionally, vitamin D is frequently added to oral nutritional supplements, due to its anti-inflammatory properties, it was administered to both groups of patients in order to avoid confounding results. Due to the relevance of the discussed points, its discussion has been enlarged in the revised version of our manuscript.
Reviewer: Critical point may be the number of subjects. Drawing any conclusions from such a small study may be difficult, if not possible.
Authors: We agree with the reviewer regarding this point was discussed in the Discussion section, despite the reduced number of participants, the results are clinically relevant and in line by previous publications.
Reviewer: Table 1. The qualitative data should be presented as a raw number and percent in the brackets the sign of “%” appears only in sex columns, the rest of data is not clear on what is representing.
Authors: Categorical data are presented in percentages and the absolute number in brackets, this information was included in the table note of the revised version of the manuscript
Reviewer: Last row of the table 1- there is an overlapping of text.
Authors: we thank the reviewer for this comment, the typing error was corrected.
Reviewer: Line 205- please explain what “self-reported core” means? I assume it refers to quality of life, but it is not clear.
Reviewer: Title of table 2- “ weeks0”
Authors: we thank the reviewer for this comment, the typing error was corrected.
Reviewer: Last raw of table 2- there is an overlapping of text.
Authors: we thank the reviewer for this comment, the typing error was corrected.
Reviewer: Table 1 ,2, 3,4. Data is such small group barely can met criteria of normal distribution and homogeneity of variance. Hence it should be presented as median and lower upper quartiles. i.e. if we consider CRP in Total baseline group from table 4 ->8.5±15.0 it means that some study subjects presented negative level of CRP. Similarly NT pro-BNP.
Authors: All data was presented in median with IQR in the revised version of our manuscript.
Reviewer: Table 4- Please compare data from Total baseline and Six-months. There is no correspondence, the numbers do not represent values of laboratory tests.
Authors: p1 refers to the comparison between all patients at baseline and after twenty-four weeks
Reviewer: Figure 5. What the stars refer to?
Authors: Legend: *:p<0.05; **:p<0.01. This information was added in the figure title of the revised version of our manuscript
- Discussion part Only lines 328-330 mention the study results. The discussion part should be a trial of assessing obtained results through the prism of the cited papers. Here, the authors do not mention their results citing only the related papers.
Authors: Results are bravely mentioned in the discussion since the objective of this section was to compare the current findings with previous findings and reports in the literature. As the reviewer suggested some results were again mentioned in the revised version of our manuscript, but not all results were specified as in the results section.
Authors: we thank the reviewer for this comment, the typing error was corrected.
Reviewer 2 Report
Comments and Suggestions for Authors
The title of this article is “Mediterranean diet and hypercaloric, hyperproteic oral supple-ments in patients with heart failure, a randomized clinical trial”. This is an interesting topic. However, there are still some areas of the article that need to be revised:
1. I did not see dietary data for the Mediterranean diet (control group) and the high-calorie, high-protein OS (intervention group) in the Materials and Methods in this paper, and although it is suggested at the end that there was no monitoring of each patient's intake, at the very least the specific types of food offered should have been listed, and this data should have been listed in this section.
2. An introduction to the Mediterranean diet and hyperproteic oral supplements (OS), as well as a brief summary of relevant research, could be modestly added to the INTRODUCTION, so that the arrangement is more conducive to the reader's understanding.
3. This article points out the impact of the Mediterranean diet and treatment with hyperproteic oral supplements (OS) on patients with heart failure. However, this article lacks certain insights and the authors need to contextualize it and give more of their own views and perspectives on the future.
4. Authors are requested to carefully check the format of the references used in the article to ensure that the references are in the required format.
Comments on the Quality of English LanguagePlease revise the English expressions in the manuscript by removing unnecessary "the" from the sentences, making sure the sentences look more concise, and replacing words that appear too often in the text. And the authors are requested to carefully check the format of the references used in the article to ensure that the references are in the required format.
Author Response
We sincerely thank the Reviewer for the constructive comments, which we found very helpful towards improving the quality of our study. Accordingly, specific changes have been made in the manuscript, based on these comments, as it is described in detail below in a point-by-point description of the changes introduced, and on how Reviewer’s concerns were addressed. Changes in the manuscript are indicated in red
REVIEWER 2
The title of this article is “Mediterranean diet and hypercaloric, hyperproteic oral supple-ments in patients with heart failure, a randomized clinical trial”. This is an interesting topic. However, there are still some areas of the article that need to be revised:
Reviewer: I did not see dietary data for the Mediterranean diet (control group) and the high-calorie, high-protein OS (intervention group) in the Materials and Methods in this paper, and although it is suggested at the end that there was no monitoring of each patient's intake, at the very least the specific types of food offered should have been listed, and this data should have been listed in this section.
Authors: we agree with the reviewer regarding this topic, this information was added in the revised version of our manuscript
- Reviewer: An introduction to the Mediterranean diet and hyperproteic oral supplements (OS), as well as a brief summary of relevant research, could be modestly added to the INTRODUCTION, so that the arrangement is more conducive to the reader's understanding.
Authors: we agree with the reviewer regarding this topic, this information was added in the revised version of our manuscript
Reviewer: This article points out the impact of the Mediterranean diet and treatment with hyperproteic oral supplements (OS) on patients with heart failure. However, this article lacks certain insights and the authors need to contextualize it and give more of their own views and perspectives on the future.
Authors: This point has been enlarged in the discussion section of the revised version of our manuscript
- Authors are requested to carefully check the format of the references used in the article to ensure that the references are in the required format.
Authors: References have been checked in the revised version of our manuscript.
Comments on the Quality of English Language
Please revise the English expressions in the manuscript by removing unnecessary "the" from the sentences, making sure the sentences look more concise, and replacing words that appear too often in the text. And the authors are requested to carefully check the format of the references used in the article to ensure that the references are in the required format.
Authors: English style and references have been checked in the revised version of our manuscript.
Round 2
Reviewer 1 Report
Comments and Suggestions for Authors
The work has been edited to the possible extent. Currently, the text requires checking - in terms of missing brackets and typos, overlapping text in tables. The work in its current form is significantly better, but still raises some doubts in terms of its scientific impact. This especially applies to the small number of respondents and the discussion of the results obtained in the study.
-
Author Response
We sincerely thank the Reviewer for the constructive comments, which we found very helpful towards improving the quality of our study. Accordingly, specific changes have been made in the manuscript, based on these comments, English has been reviewed by a native English speaker and the description of the limitations of the study were enlarged. Changes in the manuscript are indicated in red